# Genome size influences plant growth and biodiversity responses to nutrient fertilization in diverse grassland communities

Joseph A. Morton [1,2], Carlos Alberto Arnillas[3], Lori Biedermann[4], Elizabeth T. Borer[5], Lars A. Brudvig[6], Yvonne M. Buckley[7], Marc W. Cadotte[8], Kendi Davies[9], Ian Donohue[7], Anne Ebeling[10], Nico Eisenhauer[11,12], Catalina Estrada[13], Sylvia Haider[14], Yann Hautier[15], Anke Jentsch[16], Holly Martinson[17], Rebecca L. McCulley[18], Xavier Raynaud[19], Christiane Roscher[11,20], Eric W. Seabloom[5], Carly J. Stevens[21], Katerina Vesela[1], Alison Wallace[22], Ilia J. Leitch[2]*, Andrew R. Leitch[1]*, Erika I. Hersch-Green [23]*

1 School of Biological and Behavioural Sciences, Queen Mary University of London, London, United Kingdom, 2 Department of Trait Diversity and Function, Royal Botanic Gardens, Kew, Richmond, Surrey, United Kingdom, 3 Department of Physical and Environmental Sciences, University of Toronto Scarborough, Toronto, Ontario, Canada, 4 Department of Ecology, Evolution, and Organismal Biology, Iowa State University, Ames, Iowa, United States of America, 5 Department of Ecology, Evolution, and Behavior, University of Minnesota, St. Paul, Minnesota, United States of America, 6 Department of Plant Biology and Program in Ecology, Evolution, and Behavior, Michigan State University, East Lansing, Michigan, United States of America, 7 School of Natural Sciences, Zoology, Trinity College Dublin, Dublin, Ireland, 8 Department of Biological Sciences, University of Toronto Scarborough, Toronto, Ontario, Canada, 9 Department of Ecology and Evolutionary Biology, University of Colorado, Boulder, Colorado, United States of America, 10 Institute of Ecology and Evolution, Friedrich Schiller University Jena, Jena, Germany, 11 German Centre for Integrative Biodiversity Research Halle-Jena- Leipzig (iDiv), Leipzig, Germany, 12 Institute of Biology, Leipzig University, Leipzig, Germany, 13 Department of Life Sciences, Imperial College London, Silwood Park, Ascot, United Kingdom, 14 Institute of Ecology, Leuphana University of Lüneburg, Lüneburg, Germany, 15 Department of Biology, Utrecht University, Utrecht, the Netherlands, 16 Department of Disturbance Ecology, Bayreuth Center of Ecology and Environmental Research, University of Bayreuth, Bayreuth, Germany, 17 Department of Biology, McDaniel College, Westminster, Maryland, United States of America, 18 Department of Plant & Soil Sciences, University of Kentucky, Lexington, Kentucky, United States of America, 19 Sorbonne Université, CNRS, IRD, INRA, Université Paris-Cité, UPEC, Institute of Ecology and Environmental Sciences, Sorbonne Université - Paris, France, 20 Department of Physiological Diversity, Helmholtz Centre for Environmental Research (UFZ), Leipzig, Germany, 21 Lancaster Environment Centre, Lancaster University, Lancaster, United Kingdom, 22 Department of Biosciences, Minnesota State University Moorhead, Minnesota, United States of America, 23 Department of Biological Sciences, Michigan Technological University, Houghton, Michigan, United States of America

* i.leitch@kew.org (IJL); a.r.leitch@qmul.ac.uk (ARL); eherschg@mtu.edu (EIH-G)

**Data Availability Statement:** The data used in this study and the R code used for data analysis are

## Abstract

Experiments comparing diploids with polyploids and in single grassland sites show that nitrogen and/or phosphorus availability influences plant growth and community composition dependent on genome size; specifically, plants with larger genomes grow faster under nutrient enrichments relative to those with smaller genomes. However, it is unknown if these effects are specific to particular site localities with speciifc plant assemblages, climates, and historical contingencies. To determine the generality of genome size-dependent growth responses to nitrogen and phosphorus fertilization, we combined genome size and species abundance data from 27 coordinated grassland nutrient addition experiments in the Nutrient Network that occur in the Northern Hemisphere across a range of climates and grassland

publically available on the Environmental Data Initiative (EDI) at: https://doi.org/10.6073/pasta/0d6b08fbcf08605881edfb7acf0a1741.

**Funding:** This work was generated using data from the Nutrient Network (http://www.nutnet.org) experiment, funded at the site-scale by individual researchers. Specifically, NE acknowledges support for the German Centre for Integrative Biodiversity Research from the German Research Foundation (DFG-FZT 118, 202548816, www.dfg.de). MWC acknowledges support from the Natural Sciences and Engineering Research Council of Canada (#386151, www.nserc-crsng.gc.ca). XR acknowledges support for CEREEP-Ecotron IleDeFrance from the "Investissements d'Avenir" program launched by the French government and implemented by ANR (ANR-11-INBS-0001 AnaEE France and ANR-10-IDEX-0001-02 PSL, anr.fr). EB and ES acknowledge support from the National Science Foundation Research Coordination Network (NSF-DEB-1042132, nsf.gov) and Long-Term Ecological Research (NSF-DEB-1234162 to Cedar Creek LTER, nsf.gov) programs, and the Institute on the Environment (DG-0001-13, environment.umn.edu). YMB acknowledges financial support to the Burren field site from a Co-Centre award (22/CC/11103, www.sfi.ie), managed by Science Foundation Ireland (SFI), Northern Ireland's Department of Agriculture, Environment and Rural Affairs (DAERA) and UK Research and Innovation (UKRI), and supported via UK's International Science Partnerships Fund (ISPF), and the Irish Government's Shared Island initiative. Genome size plant sample collection and flow cytometry from North America sites and all cell size data collection was funded by an NSF CAREER grant awarded to EHG (NSF-DEB-Award #1941309, nsf.gov). JM's PhD studentship was funded by Queen Mary University of London (www.qmul.ac.uk). The funders had no role in study design, data collection and analysis, decision to publish, or preparation of the manuscript.

**Competing interests:** The authors have declared that no competing interests exist.

**Abbreviations:** CI, credible interval; cwGS, cover-weighted genome size; GS, genome size; LRR, log response ratio; MAP, mean annual precipitation; MAT, annual mean temperature.

communities. We found that after nitrogen treatment, species with larger genomes generally increased more in cover compared to those with smaller genomes, potentially due to a release from nutrient limitation. Responses were strongest for $C_3$ grasses and in less seasonal, low precipitation environments, indicating that genome size effects on water-use-efficiency modulates genome size–nutrient interactions. Cumulatively, the data suggest that genome size is informative and improves predictions of species' success in grassland communities.

## Introduction

Genome size (GS) varies >2,400-fold across angiosperms [1], the largest range found in any comparable eukaryotic group. It has been proposed that GS variation impacts many aspects of a plant's biology, including its life cycle, nutrient demands, water-use efficiency, and minimum cell size [2]. Considering nutrient demands, species with comparatively larger GS are hypothesized to be more growth-limited by low nitrogen (N) and/or phosphorus (P) availability, and to show greater positive growth responses following N and P additions than plants with comparatively smaller GS, owing to the hypothesized increased N and P costs of building and maintaining larger genomes [3,4]. In support of these hypotheses, greenhouse experiments comparing diploid and polyploid cytotypes have shown that plants with comparatively larger genomes exhibit faster growth relative to those with smaller genomes when grown under plentiful N- and P, but such advantages are lost when either of these nutrients are limiting [5–7]. In addition, experiments at single grassland sites have shown that plant species with larger GS are more productive on N and/or P-fertilized plots compared with low N and/or P plots [3,4,8,9]. What is not known to our knowledge is how widespread interactions between GS and nutrients on plant growth are across areas in separate geographical regions, characterized by different climatic conditions and species assemblages. Grassland ecosystems cover approximately 40% of global land area, providing diverse ecosystem services [10–12], and improved understanding of GS-nutrient interactions may lend increased predictive power in terms of how these ecosystems may change following anthropogenic eutrophication under climate change.

In grasslands worldwide, N and P fertilization has been shown to increase community productivity but lower species diversity [10,11]. This is because fertilization shifts communities towards those that compete more for light than for nutrients, with the most successful plants being those that are taller, which receive more light per unit size and are able to shade out competitors [12–14]. Furthermore, research has shown that when species compete for limiting resources (such as nutrients or water), those requiring lower levels of that resource are better able to outcompete other species [10,15]. Applied to grasslands, where N and/or P availability often limit productivity [16–18], we predict that species with smaller genomes exhibit faster growth rates under ambient site conditions relative to those with larger genomes, because they have lower cellular N and P requirements. Upon fertilization, however, we predict species with comparatively larger genomes are released from GS-nutrient constraints, enabling faster growth. In part, this faster growth may arise from increased rates of cell expansion, due to their increased minimum cell size [19,20] and/or from hybrid vigor in polyploids [21]; both allowing larger GS plants to outcompete smaller GS plants via shading [8].

Climatic factors, such as temperature and water availability, could also alter the effects of nutrient limitation on productivity [16,22] and differentially influence N and P treatment responses dependent upon species' GS. For example, higher ambient temperatures favor

growth by cell division rather than by cell expansion [19] resulting in faster biochemical reaction rates [23], which could potentially increase N and P demands and cellular N and P allocation trade-offs. Warmer climates would therefore be predicted to favor smaller GS species [24,25], which have lower N and P demands and faster cell division rates [26]. Low water availability may also influence plant growth rates dependent upon GS, although opposing hypotheses exist as to whether increased water availability will favor or disadvantage species with comparatively larger GS [27,28]. For example, increased stomatal size [29] of species with larger genomes could result in increased water loss [30] and lower water-use efficiency, if the increased size leads to increased overall stomatal pore area (area × density). Under such a scenario, low water availability should favor smaller GS species independent of nutrient availability [2,28]. Alternatively, the increased cell size of larger GS species may increase their water tissue storage capacity [27], and increase water-use efficiency if the total pore area per unit of leaf area does not increase [31], enabling them to store and conserve more water and hence maintain faster growth rates under drier conditions than smaller GS species. Furthermore, intra-annual fluctuations in temperature and precipitation could also affect GS-dependent growth nutrient interactions. Areas with more restricted growing seasons or with extreme wet and dry cycles might favor faster-growing species, which may benefit species with smaller genomes that have faster cell division rates and generation times [26,28]. In contrast, areas with longer, cooler growing seasons might favor growth by cell expansion [19] and thus species with larger GS, which have greater minimum cell sizes and can potentially grow while undergoing fewer costly cell cycles than those with smaller genomes.

Plant groups often differ in their resource requirements and allocation strategies [32,33] and such differences could also influence GS-dependent growth responses to nutrients. For example, $C_4$ plants are likely to respond less to N fertilization than $C_3$ plants, as they have a higher N-use efficiency [34,35]. Therefore, $C_3$ plants with larger genomes may be more sensitive to changes in nutrient availability than $C_4$ plants. Furthermore, it might be particularily advantageous for annual species growing in more seasonal climates with extreme fluctuations in climatic conditions to possess a small genome, as it would enable them to undergo faster cell cycles and grow quicker during the short periods when conditions are favorable for growth [36,37]. Lastly, rhizobium symbiosis in legumes and the ability of geophytes to store nutrients in underground storage organs [38,39] may increase the tolerance of species with larger genomes to N and/or P limitation [40], rendering them less responsive to N and P fertilization.

To decipher how these different factors might impact the GS-dependent growth responses to N and P, we address how GS and N and P fertilization affect plant growth on 27 grassland sites distributed across 2 continents in the Northern Hemisphere (**S1 Fig** and **S1 Table**). Sites varied in both climatic conditions (e.g., temperature, water availability, and seasonality) and in species assemblages (including species differing in their photosynthetic pathway (i.e., $C_3$/$C_4$) and functional groups). All communities were on sites within the Nutrient Network, a global research collaboration that established the same experimental design which controls for nutrient treatment and which generates data on grassland productivity, diversity, and community composition (https://nutnet.org/ [41]). By combining species GS, percent cover, functional group, and site climatic data, we tested the following 3 hypotheses using a range of approaches, including phylogenetically corrected models:

**Hypothesis 1**: N and/or P fertilization reduces cover of smaller GS species and increases cover of larger GS species across a diverse range of grassland communities.

**Hypothesis 2:** The magnitude and direction of GS-dependent responses to N and/or P fertilization depends upon temperature, water availability, and seasonality.

**Hypothesis 3**: The magnitude and direction of GS-dependent responses to N and/or P fertilization varies with plant functional group and photosynthetic pathway, being more prominent in grasses than in legumes.

## Results

### GS diversity and percent cover varies between functional groups

In total, 597 individual GS values (expressed as pg per 1C, the DNA amount of an unreplicated gametic nucleus [42]) were obtained from either the Plant DNA C-values database [43] or from new flow cytometry estimates using fresh leaf material (see **Methods** and **S1 Data**). This provided GS data for 469 of the 705 species found on the 27 sites (including different site-specific values for species found across multiple sites), accounting on average for 80% of angiosperm species encountered at a site (**S1 Table** and **S2 Data**). GS ranged 230-fold from 0.21 pg/1C in *Verbascum thapsus* (Scrophulariaceae) to 48.71 pg/1C for *Sisyrinchium campestre* (Iridaceae) and displayed a positively skewed distribution, with the mode being smaller than the mean (**Figs 1A and 2**), mean GS = 2.73 pg/1C, median GS = 1.49 pg/1C, mode = 0.90 pg/1C. Significant GS differences were observed between plant functional groups ($F_{6, 418} = 4.50$, $p < 0.001$), with geophytes and $C_3$ grasses having higher mean GS than other groups (**Fig 1A**). $C_4$ grasses had a significantly lower mean GS and a more positively skewed GS distribution than $C_3$ grasses (**S3 Fig**). Across all sites, functional groups also significantly differed in plant coverage on pretreatment plots ($F_{6, 4451} = 427.4$, $p < 0.001$), with grasses and perennial forbs being the most abundant (**Fig 1B**).

### N fertilization increases community cover-weighted GS across all sites, but has a stronger effect on less seasonal or drier sites

To examine the effect of N and P fertilization on the average GS of plants growing on the experimental plots and account for differences in dominance of each species, GS and percent cover data were used to calculate a mean cover-weighted GS (cwGS) for each plot at each site. Log response ratios (LRRs) were calculated to assess: (i) the difference in cwGS between control and nutrient-treated plots ($\Delta$cwGS $_{control\ vs\ treatment}$); and (ii) the change of each plot from pretreatment conditions ($\Delta$cwGS $_{pretreatment\ vs.\ treatment}$) (see Methods).

Compared to control plots, cwGS was significantly larger on plots treated with N, both alone and in combination with P (**Fig 3** and **S2A and S2B Table**; $\Delta$cwGS $_{control\ vs.\ treatment}$: $F_{1, 592} = 18.96$, $p < 0.001$, $R^2 = 0.069$), but not on plots treated with P alone (**Fig 3** and **S2A and S2B Table**; $F_{1, 592} = 1.20$, $p = 0.274$). These increases in cwGS were partly influenced by the recruitment of new species to the sites, but mostly they were driven by the increased growth of established, large GS species, such as *Arrhenatherum elatius* (GS = 8.1 pg / 1C at Heronsbrook), *Elymus repens* (GS = 11.8 and 11.5 pg / 1C at Jena and Cedar Creek), and *Bromus inermis* (GS = 11.4 pg / 1C at Kellogg). While combined N+P-treated plots appeared to have a higher cwGS than plots treated with N alone (**Fig 3**), the interaction effect was not significant (**S2A and S2B Table**; $F_{1, 591} = 1.21$, $p = 0.271$). Similar results were observed when examining the change in cwGS over time from pretreatment conditions (**S4** Fig and **S2C and S2D Table**).

As growth and fitness responses to nutrients are likely to be influenced by temperature and water availability, the models were amended to include 4 climatic variables (annual mean temperature (MAT), mean annual precipitation (MAP), temperature seasonality, and precipitation seasonality). The inclusion of these 4 climatic variables increased the proportion of variation in $\Delta$cwGS$_{control\ vs.\ treatment}$ explained by the model (marginal $R^2$ increased from 0.069

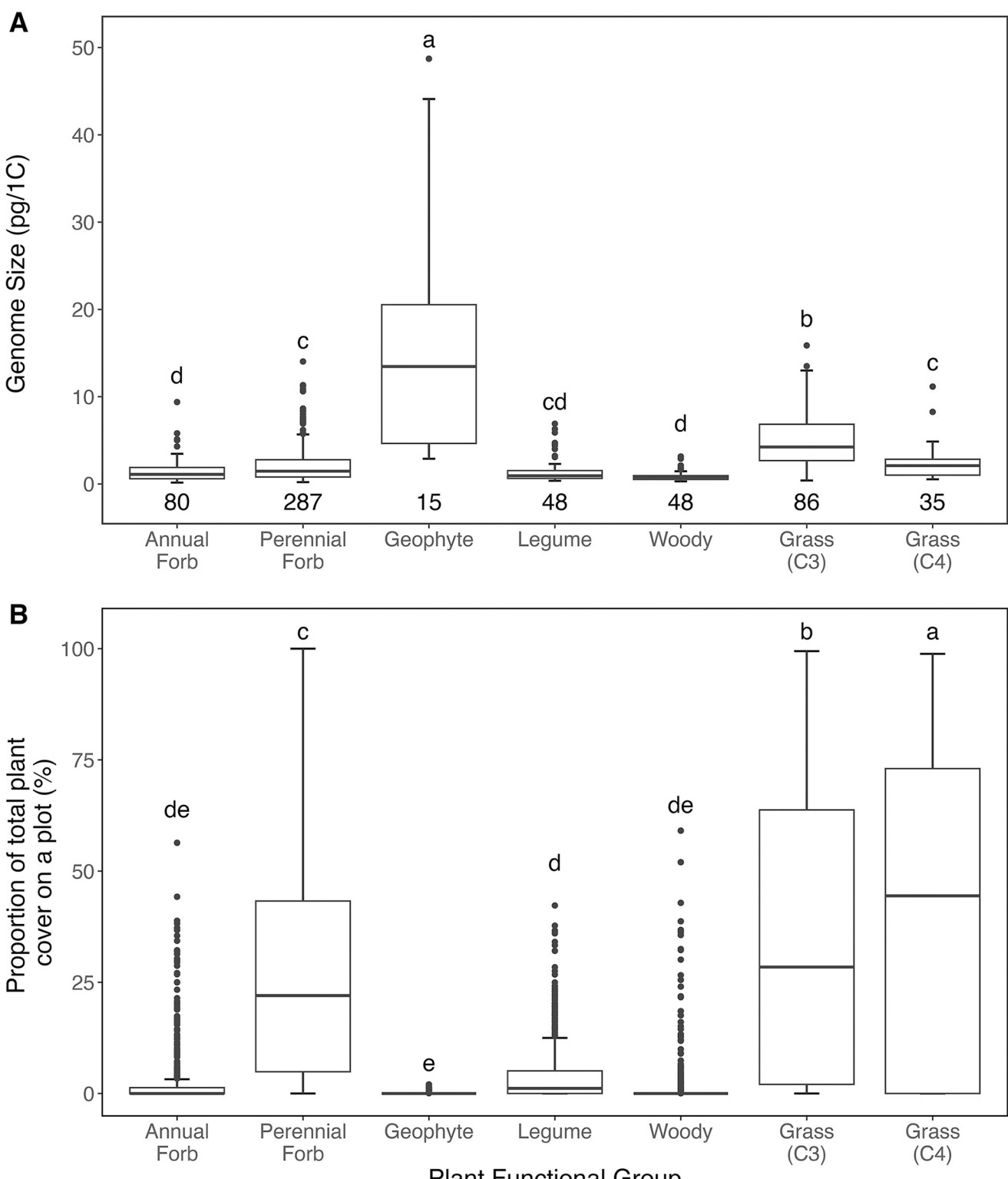

**Fig 1. Functional groups differ in GS distributions and cover on grassland plots.** (A) GS distribution of plant functional groups found on plots across the 27 Nutrient Network sites studied, with grasses split by photosynthetic pathway. GS are reported as 1C-values (pg) and were either obtained from the Plant DNA C-values database [43] or estimated from field-collected samples by flow cytometry (see Methods). The number of GS values (including multiple values for some species) is given below each box-plot. Functional groups with significantly different (Tukey's HSD test, $p < 0.05$) mean GS values are indicated by different letters. The data underlying this figure can be found in S2 Data and at https://doi.org/10.6073/pasta/

[0d6b08fbcf08605881edfb7acf0a1741](). (B) The differences in percent cover of plant functional groups found on plots across the 27 Nutrient Network sites studied, with grasses split by photosynthetic pathway. The proportion of total plant cover on pretreatment plots ($N = 730$) of all 469 species in each of 6 functional groups. Functional groups with significantly different (Tukey's HSD test, $p < 0.05$) mean cover values are indicated by different letters. The data underlying this figure can be found at https://doi.org/10.6073/pasta/0d6b08fbcf08605881edfb7acf0a1741. GS, genome size.

to 0.140) but did not alter the significance of the effect of N fertilization on ΔcwGS (**Fig 4** and **Tables 1 and S2E**). However, the difference in cwGS between control and N-fertilized plots was less on sites with higher precipitation ($F_{1, 583} = 8.87$, $p = 0.003$) and higher temperature seasonality ($F_{1, 577} = 4.25$, $p = 0.04$). ΔcwGS was not influenced by temperature or precipitation seasonality (**Fig 4** and **Tables 1 and S2E**).

## When accounting for phylogenetic history, C₃ grasses with larger GS responded most to N fertilization

While plot-level measures of GS can provide an overall summary of the effect of nutrients on the dominance of species with larger versus smaller genomes on each plot, such changes may be driven by shifts in community composition, independent of GS. Moreover, plot-level measurements do not account for differences in GS between clades, which arise as a consequence of shared evolutionary histories and patterns of ancient or recent whole genome multiplications. Indeed, a significant phylogenetic signal was observed in GS across all species (Pagel's lambda = 0.878, $p < 0.001$; Blomberg's K = $9.86 \times 10^{-6}$, $p = 0.001$; $n = 439$), suggesting phylogenetic dependence of species GS.

To account for GS and phylogenetic dependence on plant growth responses to N and P fertilization, we fitted the change in individual species percent cover from pretreatment conditions (Δcover) against GS and N and P treatment in a Bayesian phylogenetic mixed-effects model. Across all sites, species with larger genomes showed a greater increase in cover from pretreatment conditions under N fertilization compared to smaller GS species (**Table 2**, log

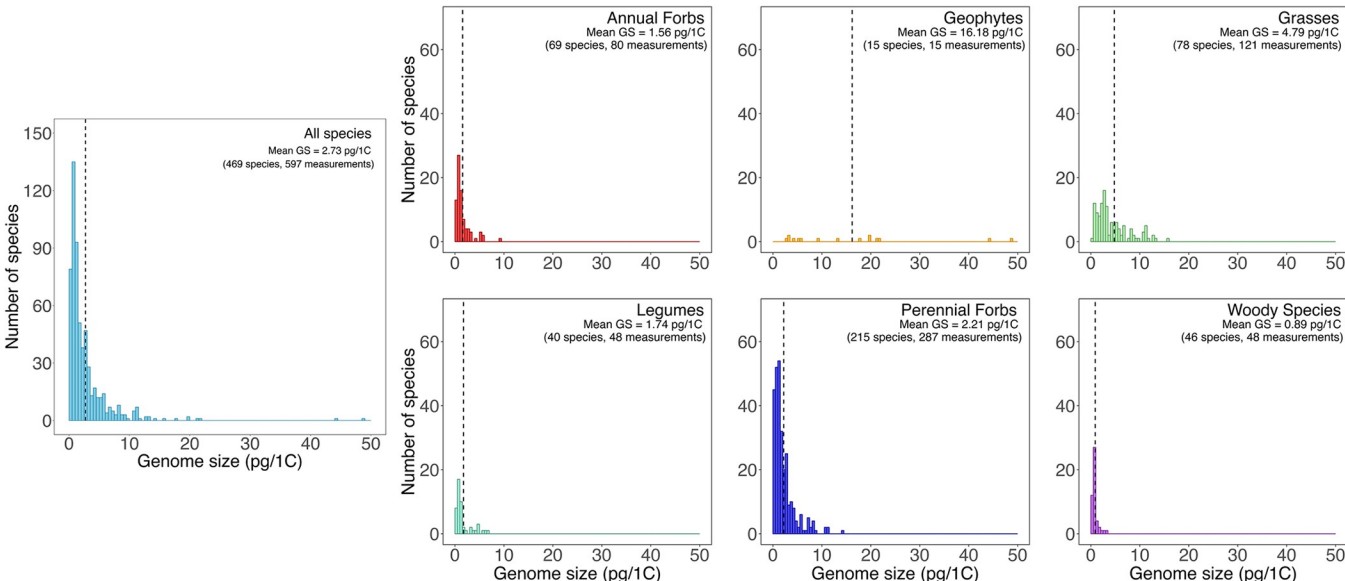

**Fig 2. GS distribution of species in the Nutrient Network sites studied mirrors that of larger databases, but varies between the 6 functional groups.** Histograms showing the GS distribution across all 469 species in the 27 Nutrient Network sites analyzed in this study, as well as across species in each of the 6 functional groups. Species count and mean GS value are given and the mean GS is indicated by a dashed line. The data underlying this figure can be found S2 Data and at https://doi.org/10.6073/pasta/0d6b08fbcf08605881edfb7acf0a1741. GS, genome size.

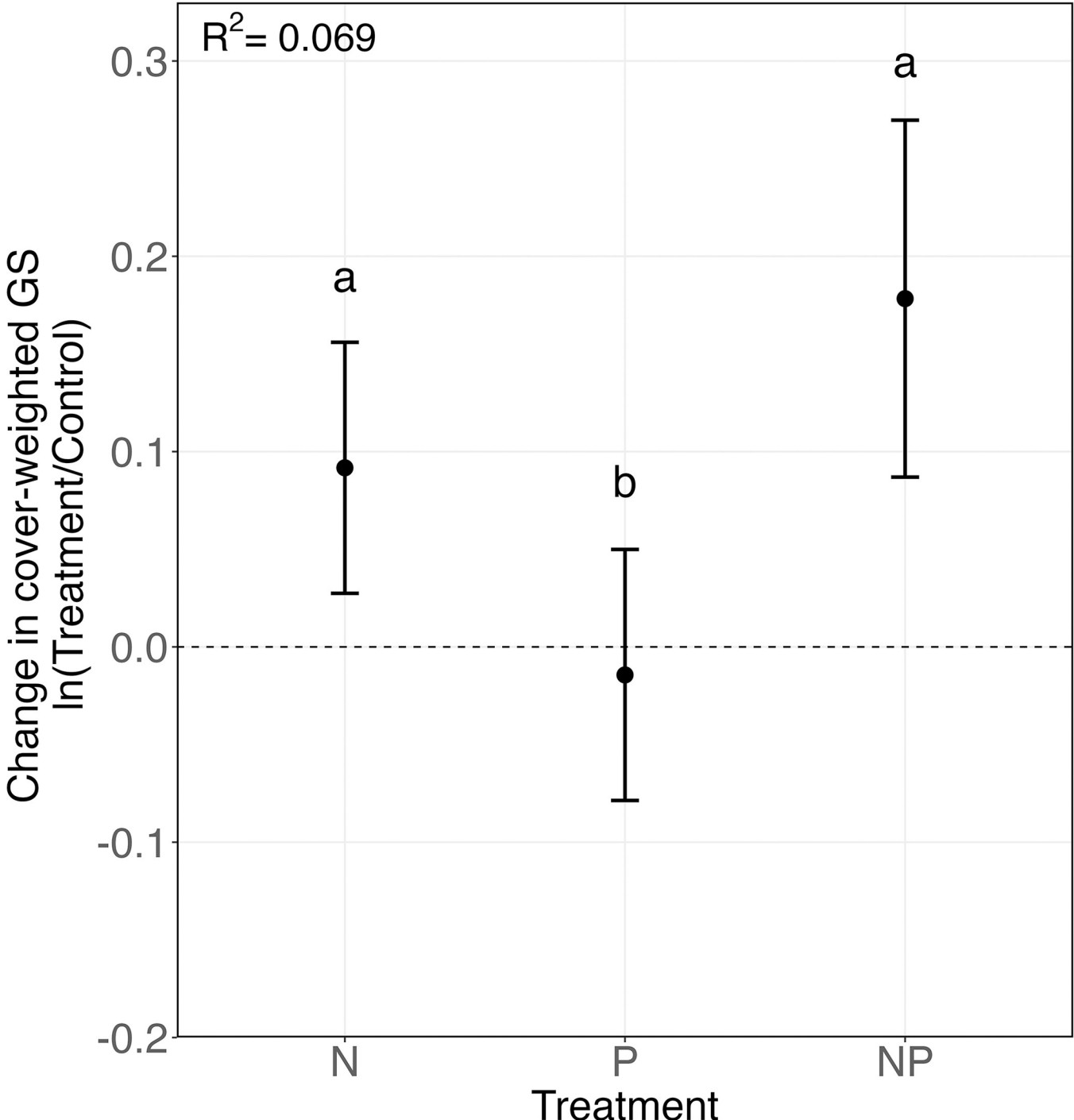

**Fig 3. Species with larger genomes become dominant on plots after nitrogen fertilization.** Average difference in cwGS between control and nutrient-fertilized conditions (indicated by dotted line) for plots with N, P, and N+P treatments. Error bars indicate 95% confidence intervals, significant differences between treatment means are indicated by letters (Tukey's HSD test $p < 0.05$) and the $R^2$ value for the fitted linear mixed-effects model is displayed ($n = 597$). The data underlying this figure can be found at: https://doi.org/10.6073/pasta/0d6b08fbcf08605881edfb7acf0a1741. cwGS, cover-weighted genome size.

(GS): N interaction in Δcover = 1.27%, 95% credible intervals (CIs) = 0.60%, 1.95%, $R^2$ = 0.168). No three-way interaction between GS, N, and P was observed (log(GS): N: P interaction in Δcover = 0.30%, CI = −0.62%, 1.21%).

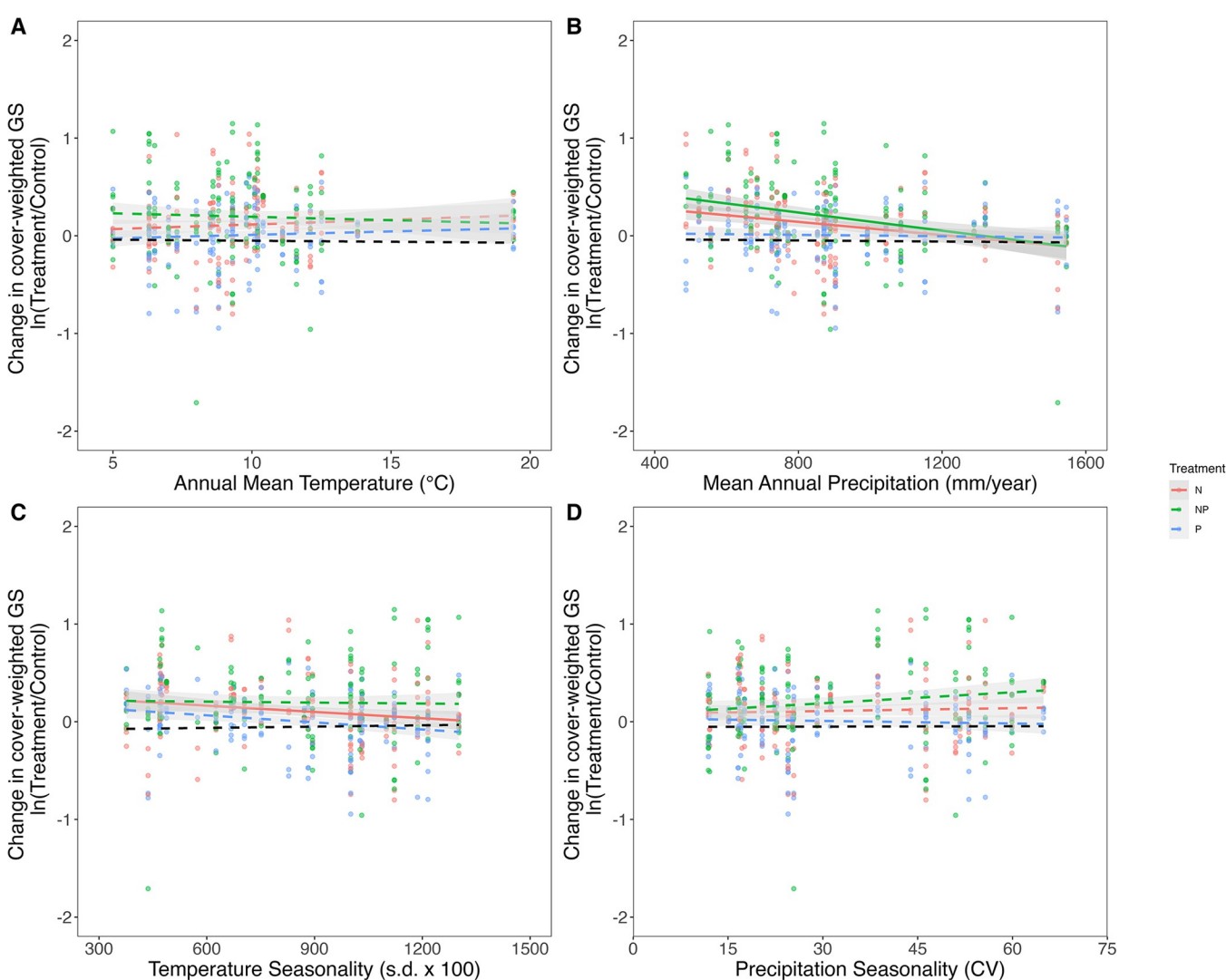

**Fig 4. The influence of GS on a plant's response to nutrient addition strengthened at lower precipitation levels and less temperature seasonality.** Average difference in cover-weighted GS ($\Delta$cwGS$_{\text{control vs. treatment}}$) from control conditions (indicated by dotted black line) for plots with N, P, and N+P treatments under varying temperature (**A**), precipitation (**B**), temperature seasonality (**C**), and precipitation seasonality (**D**). Solid lines indicate significant GS-nutrient-climate interactions, dashed lines indicate relationships that are not significant, and the gray region surrounding each line indicates 95% confidence intervals ($n$ = 597). The data underlying this figure can be found at https://doi.org/10.6073/pasta/0d6b08fbcf08605881edfb7acf0a1741. cwGS, cover-weighted genome size; GS, genome size.

Next, we examined whether different plant functional groups responded differently to N and P fertilization, by including functional group (geophyte, grass, legume, woody, annual forb, or perennial forb) as an interaction term in the phylogenetic mixed-effects model. An overall increased response of larger GS species to N across all species was still observed (**S3A Table**). However, larger GS grasses showed the most prominent increase in cover with N fertilization (**S3A Table**; log(GS): N: grass interaction in $\Delta$cover = 1.37%, CI = 0.13%, 2.60%). When grasses were split by photosynthetic pathway, only C$_3$ grasses (not C$_4$ grasses) with larger genomes showed a greater increase in cover on N-fertilized plots (**S3B Table**; $\Delta$cover of C$_3$ grasses = 1.49%, CI = 0.34%, 2.63%; $\Delta$cover of C$_4$ grasses = −2.61%, CI = −0.93%, −4.26%). These results indicate that the most prominent changes in percent cover of species after nutrient fertilization are occurring in large GS C$_3$ grasses.

**Table 1. The influence of GS on a plant's response to nutrient addition is climate dependent.** Linear mixed-effects model fitting the effects of nutrients, temperature, and precipitation on the LRR of difference in cover-weighted GS ($\Delta$cwGS $_{control\ vs.\ treatment}$). Significant differences are shown in bold and starred (* = $p \leq 0.05$, ** = $p \leq 0.01$, *** = $p \leq 0.001$). $R^2$ = 0.14.

| Log response ratio of cover-weighted genome size ~ | Sum Sq | Mean Sq | df | F-value | *p*-value | |
|---|---|---|---|---|---|---|
| **N added** | **1.74** | **1.74** | **1, 579** | **19.78** | **<0.001** | *** |
| P added | 0.11 | 0.11 | 1, 579 | 1.26 | 0.263 | |
| N added: P added | 0.11 | 0.11 | 1, 579 | 1.23 | 0.268 | |
| Mean Temperature | 0.00 | 0.00 | 1, 55 | 0.01 | 0.939 | |
| Mean Precipitation | 0.00 | 0.00 | 1, 69 | 0.03 | 0.854 | |
| Temperature Seasonality | 0.02 | 0.02 | 1, 58 | 0.21 | 0.651 | |
| Precipitation Seasonality | 0.00 | 0.00 | 1, 59 | 0.05 | 0.825 | |
| N: Temperature | 0.01 | 0.01 | 1, 578 | 0.07 | 0.790 | |
| P: Temperature | 0.00 | 0.00 | 1, 578 | 0.04 | 0.851 | |
| N and P: Temperature | 0.02 | 0.02 | 1, 580 | 0.28 | 0.599 | |
| **N: Precipitation** | **0.78** | **0.78** | **1, 583** | **8.87** | **0.003** | ** |
| P: Precipitation | 0.03 | 0.03 | 1, 583 | 0.33 | 0.564 | |
| N and P: Precipitation | 0.00 | 0.00 | 1, 580 | 0.01 | 0.941 | |
| **N: Temp. Seasonality** | **0.37** | **0.37** | **1, 577** | **4.25** | **0.040** | * |
| P: Temp. Seasonality | 0.31 | 0.31 | 1, 577 | 3.52 | 0.061 | |
| N and P: Temp. Seasonality | 0.30 | 0.30 | 1, 577 | 3.45 | 0.064 | |
| N: Precip. Seasonality | 0.00 | 0.00 | 1, 583 | 0.02 | 0.883 | |
| P: Precip. Seasonality | 0.00 | 0.00 | 1, 583 | 0.02 | 0.893 | |
| N and P: Precip. Seasonality | 0.02 | 0.02 | 1, 581 | 0.18 | 0.670 | |

cwGS, cover-weighted genome size; GS, genome size; LRR, log response ratio.

## Discussion

### N fertilization favors species with larger genomes across diverse grasslands

Both diploid-polyploid comparisons and single-location experiments have observed that nutrient fertilizations result in relatively greater growth of larger GS species [3,4,8,9]. Yet, to our knowledge, no study has tested the generality of these trends across broad environmental gradients and species assemblages. Our analysis of 27 sites, testing hypothesis 1 that "N and/or P fertilization reduces cover of smaller GS species and increases cover of larger GS species across

**Table 2. Outputs of Bayesian species-level phylogenetic mixed-effects model for all species.** Phylogenetic mixed-effects models were fitted in brms [44] to examine the effect of GS on the change in a species' percent cover with N and/or P fertilization from pretreatment conditions and the interaction between the two. Intercept (no nutrients added) and slope values are given in the table, with standard error and effective sample size. Interactions showing a slope with nonzero 95% CIs are highlighted in bold ($n$ = 439, $R^2$ = 0.168).

| Change in % cover ~ | Estimate | Standard error | CIs (95%) | Effective sample size |
|---|---|---|---|---|
| No nutrients added | −0.04 | 4.68 | −9.28, 9.65 | 12,500 |
| N added | −0.64 | 0.39 | −1.40, 0.13 | 25,024 |
| P added | 0.03 | 0.39 | −0.75, 0.80 | 24,826 |
| N added: P added | −0.68 | 0.52 | −1.68, 0.33 | 25,533 |
| log(GS) | 0.26 | 0.46 | −0.65, 1.17 | 25,952 |
| **log(GS): N** | **1.27** | **0.35** | **0.60, 1.95** | **23,989** |
| log(GS): P | 0.62 | 0.35 | −0.06, 1.30 | 24,069 |
| log(GS): N: P | 0.30 | 0.47 | −0.62, 1.21 | 26,145 |

CI, credible interval; GS, genome size.

a diverse range of grassland communities," showed that N fertilization did result in increases in the percent cover of species with large genomes. This occurred at both the plot and individual species levels of analysis (**Fig 3** and **Table 2**), though plant GS-dependent growth responses also varied according to climatic conditions and species functional attributes. These results build on single-location experiments conducted previously and suggest that increased N costs are associated with larger GS, even in complex natural systems with different biotic (e.g., mycorrhizal assemblages) and abiotic (e.g., water availability and temperature) factors.

While the effect of GS on community responses to fertilization presented here is small, it is remarkable that this signal is apparent across such a broad range of species assemblages and climatic conditions and despite the use of some conservative estimates of GS taken from the Plant DNA C-values database. When the data were analyzed using only the data for which we had directly measured GS and only on sites where we had collected samples, the effect of N fertilization on both cwGS and the change in cover of larger genomes became more prominent (**S2F–S2I** and **S3C** **Tables**). This suggests that some noise in the data is due to our choice to be conservative in GS estimates (choosing the smallest value), which may have biaised the data towards smaller GS values in species with polyploid cytotypes [45]. These results are also of similar magnitude to the effect sizes observed for genome size-nutrient interactions in single-site studies [3,4,46]. Furthermore, a post hoc analysis also revealed that the effect of N on cwGS strengthened the longer a plot had been treated with nutrients (**S4A and S4B Table**). This may be due to newer sites experiencing greater fluctuations in community composition in the first years after treatment commences (which would add noise to the data), as has been observed elsewhere [8].

The results suggest that N fertilization removes growth restrictions on larger GS species, enabling those species, which exhibited slower rates of growth under lower N conditions, to increase biomass and outcompete smaller GS species. This corroborates a recent grassland field experiment in Inner Mongolia, which after just 3 years of fertilizer treatment, found that N and P addition resulted in more rapid biomass production of larger GS species compared with smaller GS species, causing the smaller GS species to be shaded out and lost from the community [8]. The increased competitiveness of larger GS species compared to smaller GS species in the presence of plentiful nutrients may be a consequence of their larger minimum cell size, which enables more rapid growth of tissue by cell expansion [19]. Indeed, a significant negative relationship was observed between leaf cell density with genome size (**S5 Fig** and **S5 Table**), a consequence of species with larger genomes having bigger cells, corroborating existing literature [29,47]. Faster growth rates may also be driven by polyploidy, which is often associated with hybrid vigor [21]. While ploidy levels were not examined here, previous pot-experiments have found that polyploid cytotypes put on more biomass than diploid cytotypes in the presence of plentiful N and P [6,7].

Potentially, species with larger genomes respond most to N and P treatment because chromatin is rich in both N and P. However, despite the importance of both N and P for building and expressing genomes, this study finds that changes in percent cover and community cwGS were most prominently observed on plots fertilized with N or N and P, but not those fertilized with P alone. This suggests that across the 27 NutNet sites studied here, variation in GS is mostly impacting a plant's responsiveness to N availability. Synergistic effects of both N and P on productivity and diversity have been observed across terrestrial and aquatic systems [17,18], including across the Nutrient Network [19]. While this study revealed no significant additional effect of combined N and P fertilization compared to N alone, we note that a significant N:P interaction was observed when only species with direct GS measurements were used (**S2F–S2I Table**), and moreover, that the effect of N fertilization on cwGS was significantly strengthened on plots with naturally higher soil P content prior to treatment (**S4C and S4D**

**Table**), suggesting co-limitation of N and P on the growth of larger GS species. Similar results were observed in the Inner Mongolia fertilization experiment, which found that the effects of N:P interactions on biomass production were small compared to the effects of N alone [8]. The weak evidence for N:P interactions observed here may be due to insufficient power and increased noise, arising from (1) the larger range of study systems included, encompassing different grassland communities; (2) the wider range of climatic conditions; and/or (3) the use of percentage cover to weight GS measures, which may bias data towards species that are emergent from the canopy and those with larger leaves and lateral spread, and underestimate the contribution of species that are not emergent or which have thinner leaves.

## The strength of GS-dependent responses to N fertilization depends upon precipitation and temperature seasonality

Our analysis of the impact of climate on plant GS-dependent cover responses supported part of our second hypothesis that the magnitude and direction of GS-dependent responses to N and/or P fertilization depends upon temperature, water availability, and seasonality. We showed that the effects of N fertilization on the percent cover of large GS species are weakened on sites with higher temperature seasonality (**Fig 4**). This may reflect the shorter growing seasons of such locations, which are thought to favour species with smaller genomes due to their shorter cell cycle lengths and thus growth potential [19,26].

More pronounced responses of larger GS species to N fertilization were also observed on drier sites characterized by lower mean annual precipitation (**Table 1**). Because minimum cell size has been shown to scale with GS [20,28], reducing cell density (**S5 Fig and S5 Table**), this may benefit species with larger genomes under low water availability, as having larger cells may enable them to conserve or store more water in larger vacuoles [27] and maintain faster growth rates through cell expansion when nutrients are added. Furthermore, lower stomatal density of larger GS species may be sufficient to offset any increased transpiration rates of larger stomata, providing further advantage to species with larger genomes under dry conditions [2,47]. Such advantages may be diminished under wetter conditions, because increased transpiration rates in species with smaller GS might facilitate greater mass flow and thus pull nutrients more effectively from the soil to increase growth rate [48,49]. Such a scenario could reduce the impact of N on the percent cover of larger GS species relative to smaller GS species. Whatever the cause, the GS of a species does appear to affect complex trade-offs between water-use efficiency and nutrient demands that impact the production of biomass and competitiveness. Such trade-offs may also be influenced by ambient temperatures and soil texture [50], although in our analyses we were unable to test the effect of temperature and soil texture on water-nutrient trade-offs.

## The strength of GS-dependent responses to N fertilization varies with functional group

As discussed in hypothesis 3 of the introduction, differences in physiological adaptations between plant functional groups, that may or may not be associated with GS, may alter growth responses to nutrient fertilization [32,33]. While species with larger genomes generally displayed a greater response to N fertilization than those with smaller genomes (**Table 2**), this response was most prominent in grasses (**S3A Table**), especially $C_3$ grasses (**S3B Table**). Grasses are thought to have increased water-use efficiency, due to their unique "dumbbell" stomatal structure [30,51], and have been found to respond more strongly to nutrient enrichments than other plant functional groups, especially in grasslands with lower precipitation [52,53]. Thus, changes in productivity of grass species with larger genomes may be driven, in

part, by nutrient enrichment enabling faster growth and reduced water loss due to their "dumbbell" stomata. Furthermore, $C_4$ grasses may be less vulnerable to nutrient limitations, resulting in reduced responsiveness to fertilization compared to $C_3$ grasses. For example, carbon-concentrating mechanisms in $C_4$ plants enable a higher photosynthetic nitrogen-use efficiency than in $C_3$ plants [34,35] and $C_4$ plants typically invest more in roots than $C_3$ plants, enabling more efficient water and nutrient acquisition from the soil [54].

Differences in other attributes such as resource allocation and/or storage strategies between plant functional groups could also contribute to their different responses to nutrient fertilization [32,33]. For example, underground storage organs reduce the sensitivity of geophyte species to low environmental nutrient availability, potentialy enabling larger GS geophytes to remain competitive with smaller GS species [40]. Furthermore, the presence of nodules in legumes that support symbioses with N-fixing *Rhizobium* bacteria and enhance nutrient availability may explain why legumes, were less responsive to N fertilization, a trend previously observed in other studies utilizing the data from sites within the Nutrient Network [55].

Lastly, the more pronounced response of grasses, especially $C_3$ grasses, compared to other functional groups could also reflect a statistical power issue. For example, out of all functional groups, grasses were the most dominant functional group at the plot level, thus providing sufficient GS variation between the species and across the sites studied to observe differences in response to fertilization (**Fig 1**). Compared to $C_3$ grasses, there were fewer $C_4$ grass species, occupying only 14 of the 27 sites and these species exhibited a relatively small range in GS as well as a smaller maximum mean GS (**Figs 1A and S3**). This may have reduced the likelihood of effects becoming apparent in C4 grasses. Similarly, the low occurrence and diversity of some functional groups (such as geophytes and woody plants) may also explain the nonsignificant response observed there too, and further study with a broader range of species within these functional groups would be needed to establish if the lack of nutrient-GS interaction effects observed in these groups is due to biological reasons.

## Conclusions

Across 27 grassland communities composed of different plant species assemblages and occuring in widely variable climatic conditions, we find that nitrogen availability alters plant community structure based on GS. Our results show that under nitrogen-enriched conditions, species with larger GS belonging to multiple functional groups (but most notably $C_3$ grasses) were more dominant than species with smaller genomes. Furthermore, the magnitude of these effects are climate dependent, with the effects of nutrient enrichment on percent cover of large GS species being more pronounced in drier climatic conditions. These data suggest that GS might be an informative character in ecological models that aim to predict the effects of eutrophication or climate change on species vulnerabilty, success or community composition.

## Methods

### Sites and experimental design

The study was conducted across 27 Nutrient Network sites (https://nutnet.org/) in Europe and North America, spanning gradients of MAP from 487 to 1,546 mm and MAT from 5.0 to 19.4˚C (**S1 Fig and S1 Table**). At each site, fertilization treatments were randomly assigned to 25 m$^2$ plots in fully factorial combinations (control, nitrogen-added, phosphorus-added, and nitrogen- and phosphorus-added), replicated across 2 to 6 blocks. N and P were applied annually before the growing season at a rate of 10 g m$^{-2}$ (except one site, CEREEP, which applied 2.5 g.m$^2$; see Borer and colleagues [41] for full experimental design) and sites varied in length of time since nutrient treatments were initiated (2 to 14 years, **S1 Table**).

## Site selection and GS data

To select sites, 1C-values were taken from the Kew Plant DNA C-values database [43] to determine what percentage of angiosperm species at each Nutrient Network site had GS data. The 27 sites chosen had at least 55% of the species at a site and at least 1 species per plot (**S1 Table**). Multiple GS values were available for 33 of the 352 species with GS data, and all indicated the presence of different ploidy levels within a species (cytotypes). In these instances, we chose the diploid value, a choice that represents the most conservative estimate of GS for those species, as it reduces the overall effect size that GS might have when the data are analyzed at the species, commununity and/or functional group levels.

To improve the representation of GS data, we also directly measured 1C-values for 183 species at 12 sites from samples collected between 2020 and 2022, using a one-step flow cytometry procedure [56,57]. Briefly, the sample and an internal standard were co-chopped in buffer, stained with propidium iodide and the nuclear DNA content was measured using either a Sysmex CyFlow Space flow cytometer (Partec GmbH, Germany) for samples from Europe or an Accuri flow cytometer (Accuri Inc Ann Arbor, Michigan, United States of America) for samples from North America. Low-quality samples (CVs of flow histogram peaks >5%) were removed prior to analysis. For species that were recorded on and collected from different sites, we accounted for potential ploidy variation by collecting unique site species-specific 1C-values (for details on values, methods, buffers, and standards see **S1 Data**) and recorded these separately. We used these site-specific values over those available in the Plant DNA C-values database.

## Percent cover and functional group data

Species coverage and richness data were calculated using data collected on plots during pretreatment and the last 3 treatment years [41]. Permanently marked 1 m$^2$ subplots were sampled annually at peak growing season, estimating the areal percent cover of each species. For each plot (including control plots), the mean percent cover for the most recent 3 years was calculated and used to calculate change in percent cover for each species (Δcover = mean percent cover–percent cover of pretreatment year, see **S1 Table** for dates of pretreament and most recent 3 years). Species functional group (geophyte, grass, legume, woody, annual forb, or perennial forb) was recorded for each species based on the classifications used by the Nutrient Network [41]; 14 sites had both $C_3$ and $C_4$ grasses, and at these sites, grasses were classified into $C_3$ or $C_4$ based on the classification of Osborne and colleagues [58]. Data processing and statistical analyses were carried out in R v.4.2.2 [59].

## Selection of climatic variables

Nineteen BioClim variables from WorldClim v.2 [60] were extracted for each site at the 30 arc second scale. Principal component analysis of 8 precipitation and 11 temperature variables (**S2 Table**) were fitted across all sites. From contributions of each climatic variable to the principal components and a priori hypotheses, MAT (BIO1), MAP (BIO12), the variability in temperature (BIO4), and the coefficient of variation of precipitation (BIO15) were chosen as measures for temperature, precipitation, temperature seasonality, and precipitation seasonality, respectively. Climate variables were then scaled by z-score standardization for use in statistical analyses, as they differed substantially in scale and magnitude of variation.

## Phylogeny and phylogenetic signal

To obtain a phylogeny of species at the 27 sites, the phytools R package [61] was used to prune an existing NutNet phylogeny, derived from the PhytoPhylo megaphylogeny [62]. This

phylogeny was further pruned to get a smaller phylogeny of the grass species found across the sites studied. The phylogenetic signal in GS (not log transformed) was measured across all species using the phytools *phylosig* function [63,64].

## Statistical analysis

Analysis of plot-level metrics described below (total cover and cover-weighted GS) was carried out in R using linear mixed-effects models, fitted using the "lme4" R-package [65], with block nested within site treated as a random effect. Models were tested using analysis of variance (ANOVA), with *p*-values being calculated using the "lmerTest" R-package [66]. For all models, diagnostic plots were used to check for non-normal distribution of residuals and heteroscedasticity, and data were transformed where necessary to ensure the assumptions of the models were met.

## Variation in GS and percent cover between functional groups

Differences in mean GS (expressed as the log-transformed 1C-value) between functional groups (geophyte, grass, legume, woody, annual forb, or perennial forb) were tested across all sites using a phylogenetic generalized least squares model, built using the pruned Nutrient Network phylogeny and the *caper* R package [67]. Differences in the proportion of total plant cover taken up by each functional group were also tested across pretreatment plots using linear mixed-effects models. When significant effects of functional group on GS and percent cover were observed, post hoc Tukey tests were performed to identify significant differences between individual functional groups.

## The effect of nutrient fertilization, temperature, and precipitation on GS, weighted by percent cover

The cover-weighted mean GS (cwGS) of each plot on each site (including control plots) was calculated using weighted least squares models from the "nlme" R package [68]. The proportion of total plant cover taken up by each species on a plot was used to weight its contribution to the "community mean GS value," such that more dominant species had a greater influence on the mean GS than less dominant species. GS was log-transformed before calculation of cwGS to account for the high positive skew in GS data. To examine the change in average plot GS with fertilizer treatment, LRRs were used to calculate the change in cwGS value (ΔcwGS) in 2 ways:

1. Change in cwGS between control and nutrient-treated (N, P, or N+P) plots, reflecting the effect of treatment on cwGS:

$$\Delta cwGS(control\ vs.treatment) = ln\left(\frac{cwGS\ of\ nutrient\ treated\ plot}{average\ cwGS\ of\ control\ plots}\right)$$

2. Change in cwGS of each plot from the cwGS of the plot before treatment commenced, reflecting changes that occurred over the course of the experiment:

$$\Delta cwGS(pretreatment\ vs.treatment) = ln\left(\frac{cwGS\ of\ plot}{cwGS\ of\ pretreatment\ plot}\right)$$

To examine changes in cwGS with N and/or P fertilization, linear mixed-effects ANOVAs were used to fit ΔcwGS (both control versus treatment and pretreatment versus treatment) against N addition, P addition and the interaction between the 2 nutrient treatments (N:P). Post hoc Tukey tests were used to identify significant differences between individual nutrient treatments. $\Delta cwGS_{(control\ vs.\ treatment)}$ was also fitted against nutrient treatment and the 4 selected climate variables, including three-way interactions between the 2 nutrient treatments (N and P) and each climate variable.

In post hoc analysis, $\Delta cwGS_{(control\ vs.\ treatment)}$ was fitted against nutrient treatment and pretreatment soil N (%) and pretreatment soil P (ppm) to examine the effect of pretreatment conditions on cwGS response, and fitted against plot age to establish if time since first treatment affected the strength of cwGS response.

## The effect of nutrient fertilization on percent cover of species with varying GS, analyzed by functional group

Change in percent cover (Δcover–see above) was fitted against GS (expressed as log-transformed 1C-value) and N and P treatment in a phylogenetic mixed-effects model, using the *brms* R package [44] and the pruned Nutrient Network phylogeny. This model was then repeated including the interactions between the GS, N, P, and plant functional group, to test for differences in response among different functional groups. Lastly, a model was run for grass species only, with photosynthetic pathway included as an additional explanatory variable, allowing separation of responses of $C_3$ and $C_4$ grasses. For all models, weak priors were used where the slope of the regression b = normal (0,1), but models were also run with a wider range of priors, to test if prior choice impacted the convergence and output of the model. Models were run with 3 chains for 15,000 iterations, with a burn in of 5,000, and the model plots were used to examine posterior distributions and multiple chain convergence.

## Cell density

To test correlations between cell density and GS, samples were collected from 63 species across 6 sites (Cedar Creek, Chichaqua Bottoms, Kellogg, Konza, Spindletop, and Temple). Cell density per $cm^2$ fresh material was measured by digestion in 100 μl of 10% chromic acid, followed by counting the number of cells in three 10 μl aliquots on a hemocytometer and averaged. The relationship between GS and cell density was tested using linear mixed-effects models as above, with site treated as a random effect.

## Supporting information

**S1 Table. List of Nutrient Network sites used in this study.** Name, location, and study period of each site is given, as well as habitat type, elevation, climate, number of replication blocks, and the proportion of species with available genome size data (GS coverage). The climate variables used are taken from WorldClim v.2 30s data: Annual mean temperature (MAT, BIO1), mean annual precipitation (MAP, BIO12), Temperature Seasonality (BIO4), and Precipitation Seasonality (BIO15).
(DOCX)

**S2 Table. ANOVA outputs of weighted genome size (GS) models.** Results of linear mixed-effects models fitting the effect of N and P fertilization on the change in cover-weighted GS, compared to control plots ($\Delta cwGS_{(control\ vs.\ treatment}$, **a, b**) and compared to pretreatment levels ($\Delta cwGS_{(pretreatment\ vs.\ treatment}$, **c, d**) (*n* = 681). In addition, the summary table is presented for a model fitting the effects of nutrients, temperature, and precipitation on the log response ratio

(LRR) of cover-weighted GS ($\Delta$cwGS(control vs. treatment)). (**e**) Models in **a–d** were also repeated including only the species for which directly measured GS values exist, only on sites where they were sampled. (**f–i**) Significant differences are shown in bold and starred (* = $p \leq 0.05$, ** = $p \leq 0.01$, *** = $p \leq 0.001$).
(DOCX)

**S3 Table. Outputs of species-level phylogenetic mixed-effect models.** Phylogenetic mixed-effects models were fitted in brms [44] to examine the effect of genome size (GS) on the change in a species' percent cover with N, P fertilization. In addition to the model outlined in **Table 2**, a model was fitted to include the interaction between the above factors and plant functional group (**a,** $n = 439$). Another model was also run for grass species only, with photosynthetic pathway include as an additional explanatory variable (**b,** $n = 72$). Lastly, a model was run as in **Table 2** but only including species for which direct GS measurements were available (**c,** $n = 172$). Estimated intercepts and slope values showing a slope with nonzero 95% credible intervals are highlighted in bold.
(DOCX)

**S4 Table. ANOVA outputs of post hoc weighted genome size (GS) models. (a, b)** The ANOVA output (**a**) and summary table (**b**) for a linear mixed-effects model fitting the effect of N and P fertilization and plot age on the change in cover-weighted GS, compared to control plots on 27 sites in the Nutrient Network ($\Delta$cwGS(control vs. treated), $n = 589$). Significant differences are shown in bold and starred (* = $p \leq 0.05$, ** = $p \leq 0.01$, *** = $p \leq 0.001$). **(c, d)** The ANOVA output (**c**) and summary table (**d**) for a linear mixed-effects model fitting the effect of N and P fertilization, pretreatment soil N (%), and pretreatment soil P (ppm) on the change in cover-weighted GS, compared to control plots on 20 sites in the Nutrient Network ($\Delta$cwGS(control vs. treated), $n = 557$). Significant differences are shown in bold and starred (* = $p \leq 0.05$, ** = $p \leq 0.01$, *** = $p \leq 0.001$).
(DOCX)

**S5 Table. ANOVA output of cell density—genome size model.** Results of a linear model showing the relationship between log-transformed genome size (GS) and cell density across 6 sites in the Nutrient Network ($n = 81$, sites: Cedar Creek, Chichaqua Bottoms, Kellogg, Konza, Spindletop, and Temple). Significant differences are shown in bold and starred (* = $p \leq 0.05$, ** = $p \leq 0.01$, *** = $p \leq 0.001$).
(DOCX)

**S1 Fig. Distribution of the 27 sites used in this study.** Sites form part of the Nutrient Network, a global collaborative network of experimental fertilized grassland field trials, and are indicated by green circles. The map was produced from a Natural Earth data shapefile (www. naturalearthdata.com) and compiled and plotted using the *rnaturalearth* and *ggplot2* R-packages.
(TIFF)

**S2 Fig. Principal component analysis (PCA) for climate variable selection.** The 20 BioClim variables (displayed above) from WorldClim v.2 were extracted for each site at the 30 arc second scale. Principal component analysis was used to identify variables that explained the largest proportion of variation in precipitation and temperature across the 27 sites studied. PCA plots split into precipitation (left) and temperature variables (right). Coordinates of sites are labeled and are colored by country. The contribution of each variable to the principal components (PC) 1 and 2 are indicated by the direction and length of the arrows. Loadings are shown in the tables, with the size and depth of color of the circles indicating the contribution

of each climatic variable to the 2 PCs. Variables that contributed most substantially (boxed variables) in PC1 (Dim.1) and PC2 (Dim.2) were chosen as proxies for the 4 climatic factors—temperature, precipitation, temperature seasonality, and precipitation seasonality. The data underlying this figure can be found at https://doi.org/10.6073/pasta/0d6b08fbcf08605881edfb7acf0a1741.
(TIFF)

**S3 Fig. $C_3$ grasses have a higher average genome size than $C_4$ grasses.** The histogram shows the distribution of genome size (GS) across $C_3$ and $C_4$ grasses, colored by photosynthetic pathway. The inlaid boxplot shows the average and range of GS of grasses for each photosynthetic-type category, with the significant difference indicated by significance stars ($p < 0.001$). $n = 86$ ($C_3$), 35 ($C_4$). The data underlying this figure can be found at https://doi.org/10.6073/pasta/0d6b08fbcf08605881edfb7acf0a1741.
(TIFF)

**S4 Fig. Species with larger genomes become more dominant on plots after fertilization with nitrogen and phosphorus.** Average cover-weighted genome size (cwGS) was calculated for plots under factorial N and P treatment, using a 3-year mean for species percentage cover. Log response ratios (LRR) of cwGS relative to pretreatment were calculated to measure temporal changes in genome size (GS) in response to fertilization. Error bars indicate 95% confidence intervals. Significant differences between treatments are indicated by letters (Tukey's HSD test $p < 0.05$) and the $R^2$ value for the fitted linear mixed-effects model fitted for this data is displayed ($n = 597$). The data underlying this figure can be found at https://doi.org/10.6073/pasta/0d6b08fbcf08605881edfb7acf0a1741.
(TIFF)

**S5 Fig. Species with larger genomes have lower cell densities than those with smaller genomes.** Cell density (per $cm^2$ fresh tissue) was compared between 63 species of varying genome size across 6 sites in the Nutrient Network (Cedar Creek, Chichaqua Bottoms, Kellogg, Konza, Spindletop, and Temple). The solid blue line indicates the significant negative relationship, with the gray region representing 95% confidence intervals. The data underlying this figure can be found at https://doi.org/10.6073/pasta/0d6b08fbcf08605881edfb7acf0a1741.
(TIFF)

**S1 Data. GS methods, standards, and buffers.** A more detailed description of the methods, standards, and buffers used in measuring plant GS.
(DOCX)

**S2 Data. GS Data.** Data on where GS measurements for each species on each site were obtained from (database vs. directly measured) and the 1C-value (in pg) and CV data from samples for which GS was directly measured by flow cytometry.
(XLSX)

## Acknowledgments

This is KBS Contribution #2399.

## Author Contributions

**Conceptualization:** Joseph A. Morton, Carlos Alberto Arnillas, Ilia J. Leitch, Andrew R. Leitch, Erika I. Hersch-Green.

**Data curation:** Joseph A. Morton, Carlos Alberto Arnillas, Lori Biedermann, Elizabeth T. Borer, Lars A. Brudvig, Yvonne M. Buckley, Marc W. Cadotte, Kendi Davies, Ian Donohue, Anne Ebeling, Nico Eisenhauer, Catalina Estrada, Sylvia Haider, Yann Hautier, Anke Jentsch, Holly Martinson, Rebecca L. McCulley, Xavier Raynaud, Christiane Roscher, Eric W. Seabloom, Carly J. Stevens, Katerina Vesela, Alison Wallace, Ilia J. Leitch, Erika I. Hersch-Green.

**Formal analysis:** Joseph A. Morton, Marc W. Cadotte, Yann Hautier.

**Funding acquisition:** Elizabeth T. Borer, Erika I. Hersch-Green.

**Investigation:** Joseph A. Morton, Ilia J. Leitch, Andrew R. Leitch, Erika I. Hersch-Green.

**Methodology:** Joseph A. Morton, Carlos Alberto Arnillas, Lori Biedermann, Lars A. Brudvig, Yvonne M. Buckley, Marc W. Cadotte, Kendi Davies, Ian Donohue, Anne Ebeling, Nico Eisenhauer, Catalina Estrada, Sylvia Haider, Yann Hautier, Anke Jentsch, Holly Martinson, Rebecca L. McCulley, Christiane Roscher, Eric W. Seabloom, Carly J. Stevens, Katerina Vesela, Alison Wallace, Erika I. Hersch-Green.

**Project administration:** Joseph A. Morton, Elizabeth T. Borer, Ilia J. Leitch, Andrew R. Leitch.

**Resources:** Lori Biedermann, Elizabeth T. Borer, Xavier Raynaud.

**Supervision:** Joseph A. Morton, Ilia J. Leitch, Andrew R. Leitch, Erika I. Hersch-Green.

**Validation:** Joseph A. Morton.

**Visualization:** Joseph A. Morton.

**Writing – original draft:** Joseph A. Morton.

**Writing – review & editing:** Joseph A. Morton, Carlos Alberto Arnillas, Lori Biedermann, Elizabeth T. Borer, Lars A. Brudvig, Yvonne M. Buckley, Marc W. Cadotte, Kendi Davies, Ian Donohue, Anne Ebeling, Nico Eisenhauer, Catalina Estrada, Sylvia Haider, Yann Hautier, Anke Jentsch, Holly Martinson, Rebecca L. McCulley, Xavier Raynaud, Christiane Roscher, Eric W. Seabloom, Carly J. Stevens, Katerina Vesela, Alison Wallace, Ilia J. Leitch, Andrew R. Leitch, Erika I. Hersch-Green.

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
