## [Editor Report · Decision Letter 0]

13 Mar 2024

Dear Dr Morton, 

Thank you for submitting your manuscript entitled "Genome size influences growth and biodiversity responses to nutrient fertilisations in diverse grassland communities" for consideration as a Research Article by PLOS Biology.

Your manuscript has now been evaluated by the PLOS Biology editorial staff, as well as by an academic editor with relevant expertise, and I'm writing to let you know that we would like to send your submission out for external peer review.

IMPORTANT: I should warn you that after some discussion with the Academic Editor, we're still not 100% persuaded of the strength of advance that your study presents, so we'll be looking for some enthusiasm from the reviewers.

ALSO IMPORTANT: We think that it would be best to reviewer your paper as a Short Report. Because your manuscript is already concise, no re-formatting is needed; however, please select the article type "Short Reports" when you upload your additional metadata (see next paragraph).

Once your full submission is complete, your paper will undergo a series of checks in preparation for peer review. After your manuscript has passed the checks it will be sent out for review. To provide the metadata for your submission, please Login to Editorial Manager (https://www.editorialmanager.com/pbiology) within two working days, i.e. by Mar 15 2024 11:59PM.

Kind regards,

Roli Roberts

Roland Roberts, PhD

Senior Editor

PLOS Biology

rroberts@plos.org

---

## [Decision Letter · Decision Letter 1]

18 May 2024

Dear Dr Morton,

Thank you for your patience while your manuscript "Genome size influences growth and biodiversity responses to nutrient fertilisations in diverse grassland communities" was peer-reviewed at PLOS Biology. It has now been evaluated by the PLOS Biology editors, an Academic Editor with relevant expertise, and by four independent reviewers. 

Reviewer #1 is positive, but raises a few concerns, including possible bias introduced by your plant response metric, and lack of soil N/P data. Reviewer #2 is also positive, noting that the study is confirmatory of “many local experiments” – s/he wants clarification regarding your genome size and plant group classification, asks for Figs to show effects of N fertilisation (a major claim) and the logGS:N interaction (note that for the Short Report format you are allowed up to 4 main Figures and unlimited supplementary information), and to revisit your hypotheses in the Discussion. Reviewer #3 is also fine with the study, but is surprised by the lack of effect of P (given the amount of P in DNA), and has a large number of questions about your methods. Reviewer #4 is positive (about the dataset), but s/he disagrees with your interpretation; s/he does not like Fig 1, asks why you distinguished C3 and C4 grasses but not annual/perennial, wants changes to Fig 2, is unimpressed by the differences in genome size shown in Fig 3, and thinks that you need cell size data and nutrient data to support some claims.

I discussed these comments with the Academic Editor, who agreed that you should be given an opportunity to address these concerns, but stressed reviewer #4's point that some cell size data could be useful to include and strengthen the paper.

Thank you for your patience while we considered your revised manuscript "Genome size influences growth and biodiversity responses to nutrient fertilisations in diverse grassland communities" for publication as a Initial Research Submission at PLOS Biology. Your revised study has been evaluated by the PLOS Biology editors, the Academic Editor [and the original reviewers -EDIT AS APPLICABLE].

In light of the reviews, which you will find at the end of this email, we would like to invite you to revise the work to thoroughly address the reviewers' reports.

As you will see below, the reviewers… [INSERT COMMENTS FROM AE / TAILORED COMMENTS. MAKE SURE TO EXPLICITELY OVERRULE REQUESTS IF APPLICABLE AND EMPHASIZE THE MOST IMPORTANT ISSUES THAT NEED ADDRESSING].

Given the extent of revision needed, we cannot make a decision about publication until we have seen the revised manuscript and your response to the reviewers' comments. Your revised manuscript is likely to be sent for further evaluation by all or a subset of the reviewers.

**IMPORTANT - SUBMITTING YOUR REVISION**

*Re-submission Checklist*

*Published Peer Review*

*PLOS Data Policy*

*Blot and Gel Data Policy*

Sincerely,

Roli Roberts

Roland Roberts, PhD

Senior Editor

PLOS Biology

rroberts@plos.org

REVIEWERS' COMMENTS:

Reviewer #1:

Comments:

The manuscript uses a northern hemisphere grassland nutrient addition experiment to test hypotheses that suggest plants with larger genomes will be more responsive (larger) following N/P enrichment in certain environments, compared with small genome taxa, based on earlier studies at single sites and using simple experiments. The strengths of the study are the use of data from multiple natural grasslands, the range of functional groups involved, incorporation of several climate parameters, and the use of plant response data collected over several years. Overall, I have no problems with the study, data analysis and interpretation. It's a useful addition to studies defining the ecological niche of taxa with large genomes, often polyploids, in natural ecosystems. I support its publication.

I offer the following general comments for consideration by the authors during revision.

* The plant response measure relies on estimates of cover which are very difficult for grasses and often ignore height gain. This may advantage species with lateral spread and larger leaves, biasing some groups. Could this be relevant?

* I could not find any data on the soil nutrient status at the sites (N/P) which could explain some of the noise in the response data. Is this not available?

* Check throughout for consistent use of plant response terms: growth, performance, fitness etc. to ensure these are accurate.

* In the cover weighted genome size determinations, were the same taxa involved in each treatment/control over time? Did the controls have a time aggregation similar to the other treatments when determining the cover weighted genome sizes? In other words did nutrient addition facilitate new incursions of large genome taxa, or just facilitate extra growth of established species?

* Do you think the non-significant response of annual forb, geophyte etc merely reflects their low occurrence values in the plots? 

* Line 304. Power issue is really "initial advantage" or a "priority effect".

* Do the authors have any explanation from their results for why natural grasslands in their study maintain a combination of both small and large genome species in N deficient soils? This suggests that large genome species can survive and reproduce at low N/P (i.e. have reasonable fitness), even if they do not achieve maximum growth.

Reviewer #2:

With 27 grassland sites from the NUTNET work, this study reported that N fertilization stimulated the growth of species with larger genome size, a finding has been reported by many local experiments. The new result from this study is that the interaction between GS and N response would be climate-dependent and plant group-dependent. 

This manuscript wants to examine the differences of larger GS plants and smaller GS plants in their responses to fertilization, and have talked much about large and small GS. Unfortunately, I did not find the definition or standard to classify different species into small and large GS plants. Such standard is the basis for the credibility of findings from this study.

Authors reported the GS values for 597 species (right?), and only GS data for 469 species were used in further analyses. Given that this study focused on the role GS in regulating plant growth in response to nutrient addition, it would be better to keep consistent for the data. If the 597 GS values are for the 469 species, it is important to clearly state how to assign more than one values to a particular species.

Authors classified all the plants into different groups, including grass, annual forb, perennial forb, legume. Is there any overlap for species in Geophyte and other group, such as grass, perennial forb, and legume? I don't think the Geophyte group is from the same classification system with grass, forb, and woody species.

The dependence of the impacts of N fertilization on cwGS on precipitation and temperature seasonality is one of the important findings from this study. Thus, a figure is needed to show the results clearly, rather than showing the interactive results in Table 1.

Similarly, for the result of significant interaction between logGS:N in Table 2. While the results showed a significant interaction between Log GS and N fertilization, that does not mean species with larger GS would show greater increase in response to N fertilization, as stated in Lines 174-176. Figure showing such a pattern is needed.

The same problems are for the results in lines 185-193. The statistic results just showed the significance, but did not show any pattern for the changes described here.

Three hypotheses were proposed in the Introduction, but were not revisited in the Discussion.

Title: 'plant growth' would be clearer than 'growth'

Lines 12-14 How about fertilization with other elements? Authors ascribed the positive response of larger genome plants to the release from nutrient limitation after nitrogen fertilization. Did nitrogen fertilization increase the availability of nutrients beyond nitrogen?

Line 28-33 Authors cited many individual studies from different sites. Are all the studies following the same standard for classifying smaller and larger genome size plants?

Line 42 NUTNET has investigated more beyond biodiversity.

Line 68 delete the comma

Line 78 I am afraid that you are talking about intra-annual fluctuations.

Line 92 Under which condition would small GS annuals might be advantageous?

Line 94 Why species with higher N-use efficiency respond less to P fertilization?

Line 143 It is not safe to say NP plots had higher GS than N plots, as there is no statistical difference as shown in Fig 3.

Line 197 'series' or 'species'? I think you used 'species' in the Abstract.

Line 243 But the work of Fay et al (2015 Nature Plants), also based on NUTNET sites, showed multiple nutrient limitation is widespread in the NUTNET grasslands. 

Line 314-315 did P addition alter plant community based on GS?

Reviewer #3:

The paper examines response of species of different nuclear genome sizes to nitrogen and phophorus addition in grassland communities, based on a large scale fertilization experiment. The whole study is sound, although some of the assumptions/hypotheses made in the introduction are questionable and may need better support (see below). I have a number of comments to the methods section, but most of them can be handled by better explanation/description of the data handling/analysis procedures. 

I would also appreciate discussion of the absence of the effect of phosphorus addition - one would expect that P-limitation may be as important as the N-limitation (given the high amount of P in DNA). Were all the sites only nitrogen limited? One would appreciate seeing also simple response in species composition to nitrogen and phosphorus addition as a background information.

Specific comments:

li 31: there references are papers dealing only with ploidy effects, not fully relevant to support the argument

li 52: this is true for water and nutrients, but is less true for light due to strong asymmetry in competition for light. See e.g. de Malach et al. Jecol 2016. 

li 58: it is true that larger-GS species may initially grow faster due to the potential of (large) cell expansion, but growth by cell expansion is limited and cannot be maintained for long as the cell volume cannot grow indefinitely. Further size increase must be attained by cell division which takes longer in large GS species and is likely to counteract the positive effect of cell expansion. One may therefore expect that the relationship may be nonlinear or even unimodal. It be worth to explore potential nonlinearity here. 

Actually, I would appreciate more info on distribution of GS in the main text (histograms, etc.), so that the reader can appreciate the range of GS values covered by the species involved in the study.

li 89: is there a good evidence that these organs store also soil-borne nutrients in addition to the stored carbon?

li 261: but larger cells would mean also larger stomata and potential higher water loss

li 326 how many sites were included in the study? There is no info in the methods; the discussion says 27, but the residual d.f. in the Table 1 (for climatic variables, which are site-defined) seem to indicate larger values. 

li 337: how representative is the Kew Plant DNA database? One can imagine that if plant of either small or large genome size are preferntially measured, this procedure may introduce bias into the data. 

li 342: were these multiple values indicative (in all cases) of polyploid series? 

li 343: "the most conservative estimate" - why the small GS size should be considered conservative? It this may introduce potential bias by falsely assigning species to diploid status

li 347: how were these species selected? To attain a target proportion at each site or plot? Or taking into account also their presence in individual treatments? 

li 361: it would be good to know how many years have elapsed between these three most recent years and the start of the treatment (i.e. for how long time the treatment has been applied). 

li 385: phylogenetic signal was calculated in log-transformed GS or not? 

li 389: the cover-weighted GS was log-transformed before calculation? If not, the high positive skew in the GS may highly distort the outcome. 

li 397: why this has not been detemrined using phylogenetic generalized least squares? Given the high phylosignal in GS it should be appropriate.

li 403: this is unclear - which patterns were in need of clarification and what clarification means in this context? 

409-410. This is unclear

Reviewer #4:

Overall, this is a very interesting dataset that can be used to further our understanding of grassland responses to fertilization. That said, I disagree with the interpretation of the data. A major focus of the interpretation of results revolves around the potential impact of genome size on cell size and the subsequent effects of variation in cell size on whole plant physiology and growth. Given that there is no cell size data presented all discussion around cell size as playing a role is purely speculative. I appreciate that the authors cite previous work showing relationships between cell size and genome size but this previous work also shows wide variation in cell size for a given genome size. Without cell size data to back up the inferences made it is hard to accept the author's interpretation of the data. 

Further, the analysis presented in the supplementary files seems to suggest that the physiological differences between C3 and C4 grasses may represent an overarching control over the observed GS response, not GS per se. In other words, GS is going along for the ride. Additionally, we know from a long history of C4 research that C4 grasses possess larger mesophyll cells than C3 grasses. What does this mean for arguments made throughout the paper that larger cells lead to advantages in growth (for example lines 57 - 58)? Should we not expect a greater positive response by C4 grasses than C3 grasses? 

Detailed comments below.

Figure 1. I did not find this figure very useful for outlining the 3 hypotheses presented in this paper. I do not have specific comments on how to improve the cartoon but my initial interpretation of Fig 1 a story of nutrient addition leading to bigger plants with big genomes that shade out smaller plants with small genomes. 

Figure 2A. I like seeing the genome size distribution across functional groups, but it is unclear to me why these specific functional groups were chosen. For example, a distinction is made between annual and perennial forbs but not between annual and perennial grasses, Why? Also, I looked through the data and there are both annual and perennial grasses found in these grassland communities. Additionally, a great deal of text is devoted to discussing differences between C3 and C4 grasses but there is not data presented in Fig 2A for these two functional groups. 

Side note - looking through the data I think Sporobolus cryptandrus is C4 and not C3, please double check this.

Fig 2B. As stated above it would be good to have a better handle on the representation of C3, C4, annual, and perennial grasses in these systems. Why are these functional groups not represented in Fig 2B? The authors emphasize that "GS-dependent growth responses to nutrients could also vary among plant groups" and spend significant time outlining that C3 and C4 are important functional distinctions to consider. The introduction even cites previous work suggesting that C3 and C4 plants respond differently to N and P fertilization. Since the authors acknowledge that physiological differences between C3 and C4 plants are important, why is this functional distinction missing from Fig. 2B and instead only highlighted in the supplementary data?

Fig 3. These are very small changes in cover-weighted GS.

Lines 204 - 208, 227 - 233: It would be good to have the nutrient and cell size data for these species when making these arguments. Without cell size and nutrient data this is all speculative.

Lines 258 - 259: Are these sites associated with a higher abundance of C4 plants in the control / pretreatment plots?

Lines 260 - 262: Again without cell size data this is speculative.

Line 282 - I think this should be Table S3A and S3B respectively. 

Lines 307 - 309: What does far fewer C4 grasses mean? Are there fewer species, smaller percent cover? As stated previously please show this data. This is necessary to better understand how the change in C4 cover influences the observed cover-weighted genome size response to nutrients shown in Fig 3.

---

## [Editor Report · Decision Letter 2]

23 Sep 2024

Dear Dr Morton,

Thank you for your patience while we considered your revised manuscript "Genome size influences plant growth and biodiversity responses to nutrient fertilisations in diverse grassland communities" for publication as a Short Reports at PLOS Biology. This revised version of your manuscript has been evaluated by the PLOS Biology editors and the Academic Editor.

Based on our Academic Editor's assessment of your revision, we are likely to accept this manuscript for publication, provided you satisfactorily address the following data and other policy-related requests.

a) Please change your Title very slightly to "Genome size influences plant growth and biodiversity responses to nutrient fertilization in diverse grassland communities" (i.e. "fertilization" should be singular and US spelling; we understand the reasons for it being plural, but it reads oddly in isolation, and the plurality is made amply clear in the Abstract).

b) Please address my Data Policy requests below; specifically, we need you to supply the numerical values underlying Figs 1AB, 2, 3, 4AB, S2, S3, S4, S5, either as a supplementary data file or as a permanent DOI’d deposition. I note that the raw data are already available in the EDI deposition, but we will need the values that directly underlie the Figs.

c) Please cite the location of the data clearly in all relevant main and supplementary Figure legends, e.g. “The data underlying this Figure can be found in S1 Data” or “The data underlying this Figure can be found in https://zenodo.org/records/XXXXXXXX

d) Please make any custom code available, either as a supplementary file or as part of your data deposition. I see that you have some code in Github, but because Github depositions can be readily changed or deleted, please make a permanent DOI’d copy (e.g. in Zenodo) and provide this URL.

We expect to receive your revised manuscript within two weeks. 

*Published Peer Review History*

*Press*

Sincerely,

Roli Roberts

Roland Roberts, PhD

Senior Editor

rroberts@plos.org

PLOS Biology

DATA POLICY:

Regardless of the method selected, please ensure that you provide the individual numerical values that underlie the summary data displayed in the following figure panels as they are essential for readers to assess your analysis and to reproduce it: Figs 1AB, 2, 3, 4AB, S2, S3, S4, S5. NOTE: the numerical data provided should include all replicates AND the way in which the plotted mean and errors were derived (it should not present only the mean/average values).

CODE POLICY

DATA NOT SHOWN?

---

## [Editor Report · Decision Letter 3]

5 Nov 2024

Dear Dr Morton,

Thank you for the submission of your revised Short Report "Genome size influences plant growth and biodiversity responses to nutrient fertilization in diverse grassland communities" for publication in PLOS Biology. On behalf of my colleagues and the Academic Editor, Andrew Tanentzap, I'm pleased to say that we can in principle accept your manuscript for publication, provided you address any remaining formatting and reporting issues. These will be detailed in an email you should receive within 2-3 business days from our colleagues in the journal operations team; no action is required from you until then. Please note that we will not be able to formally accept your manuscript and schedule it for publication until you have completed any requested changes.

I'd also like to add an extra reminder to finalise your EDI deposition and update the URL/DOI at the soonest opportunity.

Sincerely, 

Roli Roberts

Senior Editor

PLOS Biology

rroberts@plos.org